# Caption generation from histopathology whole-slide images using pre-trained transformers

**Bryan Cardenas Guevara**[1]                    BRYAN.CARDENASGUEVARA@SURF.NL
[1] *SURF, Amsterdam, The Netherlands*
**Niccolò Marini**[2]                             NICCOLO.MARINI@HEVS.CH
[2] *University of Applied Sciences Western Switzerland, Sierre (HES-SO Valais)*
**Stefano Marchesin**[3]                          STEFANO.MARCHESIN@UNIPD.IT
[3] *University of Padua, Padua, Italy*

**Witali Aswolinskiy**[4]                         WITALI.ASWOLINSKIY@RADBOUDUMC.NL
[4] *Radboud University Medical Center, Nijmegen, The Netherlands*

**Robert-Jan Schlimbach**[1]                      ROBERT-JAN.SCHLIMBACH@SURF.NL
**Damian Podareanu**[1]                           DAMIAN.PODAREANU@SURF.NL
**Francesco Ciompi**[4]                           FRANCESCO.CIOMPI@RADBOUDUMC.NL

**Editors:** Under Review for MIDL 2023

## Abstract

The recent advent of *foundation models* and *large language models* has enabled scientists to leverage large-scale knowledge of pretrained (vision) transformers and efficiently tailor it to downstream tasks. This technology can potentially automate multiple aspects of cancer diagnosis in digital pathology, from whole-slide image classification to generating pathology reports while training with pairs of images and text from the diagnostic conclusion. In this work, we orchestrate a set of weakly-supervised transformer-based models with a first aim to address both whole-slide image classification and captioning, addressing the automatic generation of the conclusion of pathology reports in the form of image *captions*. We report our first results on a multicentric multilingual dataset of colon polyps and biopsies. We achieve high diagnostic accuracy with no supervision and cheap computational adaptation.

**Keywords:** Whole slide images, histopathology, multi-modal training, caption generation

## 1. Introduction

Recent advances in the field of deep learning are showing increasing capability of bridging the gap between *language* understanding and *vision*. Such technology is particularly suited for the field of medical imaging, where *multimodal* data with pairs of images and text from electronic health records are clinically available. With the adoption of digital pathology workflows, an increasing amount of gigapixel whole-slide images is produced clinically, containing a wealth of information for deep learning development. However, the promise of computer algorithms as a support for pathology diagnosis and potentially aiding the generation of pathology reports often relies on supervised learning, presenting a challenge due to the substantial amount of labeled data required, together with the time-consuming interpretation of histopathology whole slide images (WSIs). The authors in (Gamper and Rajpoot, 2021) provide evidence that models pre-trained on digital pathology images learn highly informative representations for caption generation. Nevertheless, their proposed method

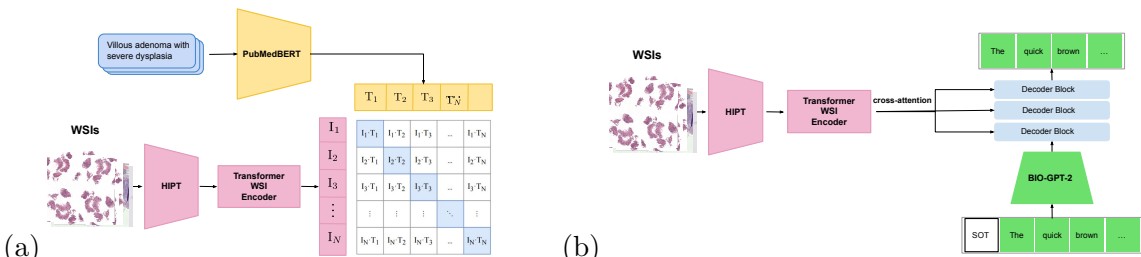

Figure 1: In the first stage (a) we perform Contrastive WSI-caption pre-training, while in (b), the decoder blocks are conditioned on the WSI embeddings trained in (a).

| Original Caption | GPT-3.5 Cleaned | Generated Caption |
|---|---|---|
| biopsies distal colon: chronic inflammation, in partially active and slightly histiocytary. no specific characteristics. the microscopic preparations from elsewhere have been requested for revision. | chronic inflammation, no specific characteristics. | no abnormalities, no dysplasia or malignancy. cyclic inflammation. |
| biopt colon transversum: adenocarcinoma. | adenocarcinoma. | Metastasis of adenocarcinoma best suited to primary process. |
| 1) fragments of tubular adenoma with high degree dysplasia. | tubular adenoma with high degree dysplasia. | adenocarcinoma on villous adenoma. no lymphovascular invasion is identified. enced enced enced ED ED ED ED |

Table 1: The last example shows a failed caption generation.

involves an exhausting effort to extract and process captions in figures from text books. Similarly, previous studies (Zhang et al., 2020; Tsuneki and Kanavati, 2022) show that caption generation is viable in digital pathology but the authors do not apply self-supervised or pre-trained models. Motivated by this, we demonstrate the benefit from fine-tuning pre-trained weakly supervised transformers on the task of pathology caption generation. We orchestrate a two-stage pipeline where we first learn highly informative image and text representations using the CLIP training regime (Radford et al., 2021)[1]. In the second stage, we utilize extracted WSI representations from the first stage to condition a pre-trained bio-gpt-2 (Luo et al., 2022) language model to generate captions. Moreover, pathology captions may include irrelevant or noisy information, such as running text unrelated to any observed lesion in the WSI. To address this, we explore the use of GPT-3.5-turbo (Ouyang et al., 2022) to pre-process the captions and remove extraneous information.

## 2. Method

**Data** We collected 5729 gigapixel-size whole slide images of colon polyps and biopsies scanned at 0.25 micron per pixel spacing originating from two labs from two countries. Each WSI-caption pair was labelled with one of five diagnostic labels: normal, hyperplasia, low-grade dysplasia, high-grade dysplasia, or adenocarcinoma. These labels were not used during training of the pipeline. A subset of 569 patient-split WSI-caption pairs was reserved for testing, which served as the basis for evaluating our results. The captions were

---

1. Our code is available on github

| Unpretrained caption model | Pre-trained caption model | GPT-3.5 cleaned pre-trained caption model | WSI supervised classifier |
|---|---|---|---|
| 0.65 (±0.20) | 0.70 (±0.21) | **0.73** (±0.15) | 0.76 (±0.16) |

Table 2: Mean F1-scores over the five diagnostic classes for each model.

machine translated (Tiedemann and Thottingal, 2020) from two languages to English. The captions were subsequently pre-processed using GPT-3.5-turbo, which was prompted with ten examples of how to restructure the captions. These processed captions are then used for training in the two-stage pipeline. Three examples are shown in Table 1.

**Architecture**  In the first stage of our pipeline, we train a CLIP model, which consists of a HIPT model (Chen et al., 2022) to encode WSIs and PubmedBERT (Gu et al., 2020) to encode medical text. The HIPT model is a hierarchical transformer model trained using DINO (Caron et al., 2021) on TCGA (Liu et al., 2018). HIPT encodes a 4096x4096 WSI region to a vector of size 192. In this manner, we extract a sequence of embeddings that represents one (packed) WSI. Subsequently, we train a transformer encoder on this sequence to pool the features and map them to the same dimensionality as the caption embeddings. We kept both the image and language pre-trained models frozen and only extract the WSI and caption embeddings. In the second stage, we extract the WSI embeddings from the previous stage and condition decoder layers on top of a pre-trained bio-gpt-2 model.

**Evaluation**  We evaluated the generated captions by assessing their diagnostic ability by manually classifying the captions to one of the five diagnostic labels. We could then compare the mean F1-score of four distinct models: (1) A supervised WSI classifier that we treat as a baseline, (2) a caption model without a pre-trained bio-gpt-2 decoder, (3) a pre-trained bio-gpt-2 caption model and (4) a pre-trained bio-gpt-2 caption model trained on the processed gpt-3.5-turbo caption data.

## 3. Results and Discussion

The application of GPT-3.5-turbo to clean the captions results in a significant improvement over the baseline models in terms of diagnostic accuracy and the quality of the generated captions. Our captioning model has a diagnostic accuracy close to a supervised classifier while having the weakly-supervised advantage. The caption templates in which the captions are written by the pathologists differ between the two labs and by prompt-style caption pre-processing we are able to normalize them. The structure of the original captions differs between the labs and by prompt-style caption pre-processing we are able to normalize them. Despite using large transformer models, we fine-tuned our pipeline on a single A100 (40GB) GPU in 20 minutes. Our work highlights the need for large scale pre-trained models in the field of digital pathology.

## Acknowledgments

This project has received funding from the European Union's Horizon 2020 research and innovation programme under grant agreement No 825292 (ExaMode, htttp://www.examode.eu/)

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
