# OpenReview forum: "Caption generation from histopathology whole-slide images using pre-trained transformers"
_MIDL.io/2023/Short_Paper_Track — MIDL 2023 Short paper track Poster_

### Official Review · Reviewer_WDoE · 2023-04-14
**great work**

**Rating:** 8
**Confidence:** 5

**Review:**

This is an excelent application of LLM and ML in the context of histopathology WSI. The authors evaluate their method on a large dataset. The results are very promising and would generate very interesting discussions. Would be great to learn if this can me extended to medical imaging domains.

---

### Official Review · Reviewer_q8HY · 2023-04-24

**Rating:** 6
**Confidence:** 3

**Review:**

# Summary

The paper applies a transformer-based caption model to generate pathology reports of whole slide images. The generated capt
ions are then used to (manually) predict a diagnosis, which is comparable in performance to a fully supervised classi
fication model.

# Strengths

I believe this is interesting work, and a well performing report generator based on WSI would be desirable. The example results of generated captions look somewhat reasonable, although only few examples were shown. I also liked the idea of evaluating the captions through of manual diagnosis based on the report.


# Weaknesses

While the first results look interesting, the experimental setup is lacking, and I could not really assess from the provided experiments and results how well the method actually works. While I like the idea of manual diagnosis based on the generated reports for evaluation, the comparison to fully supervised WSI classification does not seem appropriate. Rather, the authors should have compared against the real captions.

The description of the experiment setup is a bit unclear, and the results are very preliminary. However, as an abstract, this may be sufficient, but I would look forward to more thorough evaluation.